# Brominated Flame Retardants in Children’s Room: Concentration, Composition, and Health Risk Assessment

**DOI:** 10.3390/ijerph18126421

**Published:** 2021-06-14

**Authors:** Douha Bannan, Nadeem Ali, Nabil A. Alhakamy, Mohamed A. Alfaleh, Waleed S. Alharbi, Muhammad Imtiaz Rashid, Nisreen Rajeh, Govindan Malarvannan

**Affiliations:** 1Faculty of Pharmacy, Pharmacy Practice, King Abdulaziz University, Jeddah 21589, Saudi Arabia; Dbannan@kau.edu.sa; 2Center of Excellence in Environmental Studies, King Abdulaziz University, Jeddah 21589, Saudi Arabia; irmaliks@gmail.com; 3Pharmaceutics Department, Faculty of Pharmacy, King Abdulaziz University, Jeddah 21589, Saudi Arabia; nalhakamy@kau.edu.sa (N.A.A.); wsmalharbi@kau.edu.sa (W.S.A.); 4Natural Products and Alternative Medicine Department, Faculty of Pharmacy, King Abdulaziz University, Jeddah 21589, Saudi Arabia; maalfaleh@kau.edu.sa; 5Department of Anatomy, Medical College, King Abdulaziz University, Jeddah 21589, Saudi Arabia; nrajeh@kau.edu.sa; 6Toxicological Center, University of Antwerp, 2610 Wilrijk, Belgium

**Keywords:** brominated flame retardants, indoor dust, PM_10_, children exposure, Saudi Arabia

## Abstract

Children spend most of their daily time indoors. Many of the items used indoors, such as furniture, electronics, textile, and children toys, are treated with chemicals to provide longevity and fulfil the safety standards. However, many chemicals added to these products are released into the environment during leaching out from the treated products. Many studies have reported brominated flame retardants (BFRs) in indoor environments; however, few have focused on environments specified for young children. In this study, paired air (PM_10_) and dust samples were collected from the rooms (*n* = 30) of Saudi children. These samples were analyzed for different congeners of polybrominated diphenyl ethers (PBDEs) and three important alternative flame retardants using gas chromatography-mass spectrometry. Decabromodiphenyl ether (BDE 209) was the most important analyzed BFR in dust and PM_10_ samples with a median value of 3150 ng/g of dust and 75 pg/m^3^. This indicates the wider application of BDE 209 has implications for its occurrence, although its use has been regulated for specified uses since 2014. Among alternative BFRs, 2-Ethylhexyl-2,3,4,5-tetrabromobenzoate (TBB), Bis(2-ethylhexyl)-3,4,5,6-tetrabromophthalate (TBPH), and 1,2-Bis(2,4,6-tribromophenoxy)ethane (BTBPE) were found with a median levels of 10, 15 and 8 ng/g of dust, respectively. However, alternative BFRs were present in <50% of the PM_10_ samples. The calculated long term and daily exposures via indoor dust and PM_10_ of Saudi children from their rooms were well below the respective reference dose (RfD) values. Nonetheless, the study highlights BDE 209 at higher levels than previously reported from household dust in Saudi Arabia. The study warrants further extensive research to estimate the different classes of chemical exposure to children from their rooms.

## 1. Introduction

Many consumer products, such as building materials, including thermal insulation boards, electric and electronic equipment, furniture foams, children’s toys, fabrics, printed circuit boards, etc., are treated with flame retardants to fulfil fire safety regulations [1,2]. These chemicals are added to the products instead of via chemically reaction; thus, they can migrate into the surrounding environment from treated products [1,2]. These chemicals are of concern because of the associated health risks such as neurodevelopmental and behavioral outcomes, endocrine disruption, and possibly carcinogenicity, especially for young children due to their rapidly developing bodies [3,4,5]. The use of commercial formulations of high volume polybrominated biphenyl ethers (PBDEs), namely Penta-, Octa- and Deca-BDE, are contained in the Stockholm Convention list of persistent organic pollutants (POPs), and their production and use are regulated/restricted [6,7]. Regulations regarding PBDEs open the market for alternative brominated and phosphate flame retardants [1,8,9]. These alternative currents using flame retardants (FRs) have been suggested to be less persistent and bioaccumulative. Yet recent studies reported otherwise, and high levels of currently used-FRs were found in air, dust, and human and animal samples from various countries [1,10,11,12].

Once these chemicals are in the environment, humans become exposed to them from different exposure routes, e.g., contaminated food, air, dust, etc. Studies have reported that involuntary inhalation, dermal contact, and contaminated dust and air intake are considered primary exposure routes for many of these chemicals [2,8,13,14]. The monitoring of the indoor environment can assess, over the long term, vulnerable exposure groups such as toddlers (hand-to-mouth contact) and young children since they spend a lot of time indoors [10,15]. Analysis of indoors dust and atmospheric suspended fine particles is significant for this age group, especially in the Middle East region. Due to challenging outdoor weather conditions, children spend most of their time indoors [10,16]. Indoor dust is considered an archive of pollution that accumulates contaminants over a long time. Due to young children’s hand-to-mouth and licking habits, they involuntarily ingest and inhale chemicals and varying amounts of dust [17,18]. Due to the lack of moisture and sunlight in the indoor environment, many contaminants do not break down and show slow degradation [17].

Nevertheless, monitoring studies are very important for the detailed insight into the spatiotemporal occurrence trends of chemicals in the changing environment and to assess the effective implementation of new regulations to control their adverse impact on the environment and the human population. Saudi Arabia has been going through rapid industrialization over the past few decades; thus, the Saudi population’s lifestyle has also changed dramatically. Studies are needed to understand the impact of changing lifestyles and changing indoor environments on health. Therefore, the current study reports the incidence of these chemicals in children’s room for the first time in Saudi Arabia. This project’s specific objectives were to study the profiling of selected BFRs in indoor air and dust within children rooms from selected Saudi households and to estimate exposure to these chemicals via dust ingestion, dermal contact and air inhalation using obtained levels of these chemicals in Environmental Protection Agency (EPA) equations.

## 2. Materials and Methods

### 2.1. Chemicals and Solvents

Analytical standards of NBFRs, namely 2-Ethylhexyl-2,3,4,5-tetrabromobenzoate (TBB), Bis (2-ethylhexyl)-3,4,5,6-tetrabromophthalate (TBPH), 1,2-Bis (2,4,6-tribromophenoxy)ethane (BTBPE), and polybrominated diphenyl ethers (PBDEs) (28, 47, 99, 100, 153, 154, 183, and 209 were purchased from AccuStandards and Sigma Aldrich. BDE 77, 128, and labelled BDE 209 were used as internal standards (ISs). All stock solutions for the analytical standards were prepared in iso-octane and toluene. Acetone, dichloromethane (DCM), *n*-hexane (*n*-Hex), and iso-octane were of analytical grade obtained from Sigma Aldrich.

### 2.2. Sampling

For this study, paired particulate material (PM_10_) and indoor dust samples were collected simultaneously from children’s rooms (N = 30) in different selected households in Jeddah, Saudi Arabia, between August and December 2019. Households with young children in a separate room were randomly selected from the general population who participated. During the sampling campaign, paired samples (dust and particulate matter (PM_10_)) were collected from each household’s children’s rooms. PM_10_ are inhalable dust particles in the air with diameters generally of 10 micrometers and smaller. Weather in Jeddah is typically dry, dusty, and hot during most of the year and indoor temperature varies between 18–25 °C, due to air conditioning, depending on household preference. Therefore, sampling from a different time of the year would not have made a big difference. As described in the literature, settled indoor dust is an essential indoor pollution archive; therefore, studying pollutants from indoor dust is essential [17,19]. A questionnaire with information about the size of the room, age and number of children sharing, type of building material used, floor type, age of the building, variety of toys (rigid plastic, soft plastics, stuffed, electrical, new, old, etc.), and cross ventilation were recorded to find possible point sources for these chemicals. All this information is provided in the Appendix A. Floor dust was collected using a vacuum cleaner. All the material used for sampling was thoroughly cleaned using a solvent to escape cross-contamination. Mesh (200 µm) was used to sieve collected dust to gather homogenized dust for the quantitative analysis. A Micro-Environmental Monitor TM air sampler was used in the children’s rooms to collect the PM_10_; before each sampling, the room was cleaned with solvent. The sampler was installed for 24 h with 10 L per min (LPM) flow. PM_10_ samples were collected using 47 mm glass fiber filter paper. The filter paper was oven-baked at 400 °C for 6 hrs and kept in desiccators until use to eliminate moisture and contamination. Microbalance was used to measure the PM_10_ levels, and then the sampled filter paper was stored at −20 °C until analysis in the individual cassette.

### 2.3. Sample Preparation and Quantitative Analysis 

A detailed description of sample preparation is provided by Ali et al. [20]. Briefly, accurately measured dust (AC filter and settled dust), typically ~75 mg, was taken. After spiking with ISs, a solvent mixture of hexane: acetone (4/1, *v/v*) was added, and then samples were extracted using ultrasonication (20 min) followed by centrifugation (3000 rpm for 10 min). The supernatant was collected in a clean tube, repeating the same extraction procedure twice with the leftover sediments. The extracts were pooled and brought to incipient dryness using a gentle stream of nitrogen. After drying, samples were resolubilized in 1 mL of the solvent mixture (hexane and acetone). These samples were cleaned further using silica BondElut (Agilent technologies, Santa Clar, CA, USA) and 10 mL solvent mixture (hexane/dichloromethane). After elution, the obtained fraction was concentrated to incipient dryness under a gentle stream of nitrogen. It was then resolubilized in 100 µL of iso-octane for gas chromatography-mass spectrometry (GC-MS) analysis. The same procedure was used for the extraction of BFRs from PM_10_.

A TSQ™ 8000 Evo triple quadrupole GC-MS/MS (Thermo Fisher Scientific, Waltham, MA, USA) was used in the selected ion monitoring (SIM) mode for quantitative analysis. A fused silica capillary column (Rxi-5silMS 15 M × 0.25 mm × 0.10 µm) was used for the separation. Samples were injected in splitless mode with a split flow of 50 mL/min, and the inlet temperature was 300 °C. Helium was used as the carrier gas at 1.2 mL/min for 18 min and at 2 mL/min for the rest of the analysis. GC oven temperature was raised from 90 °C to 200 °C at 15 °C/min and then to 300 °C at 7 °C/min and hold for 7 min. The MS transfer line and ion source temperature were set at 290 °C and 250 °C, respectively. For BDE, 28, 47, 77, 99, 100, 128, 153, 154, 183 *m/z* 159, 161, and 163 were monitored. For TBB (*m/z* 357, 359), BTBPE (*m/z* 251, 253), TBPH (*m/z* 384, 386, 515), BDE 209 and its internal standard *m/z* 409, 441, 485, and 487 were monitored. 

### 2.4. Quality Assurance & Quality Control (QA/QC)

All the glassware used was baked at 400 °C overnight and kept at 100 °C till use. Standard reference material (SRM) 2585 from the National Institute of Standards & Technology (NIST), procedural blanks (one for every eight samples), and washed Na_2_SO_4_ (dust replica) spiked with a known concentration standard were used to evaluate the procedure accuracy. The levels of the analytes found in procedural blanks were corrected from the concentrations of the analysts in the samples. The experimental procedure was performed under a fume hood without light and using amber glassware to avoid photo-degradation.

### 2.5. Human Risk Assessment Calculations 

Health risk assessment to children for selected BFRs from their rooms was calculated by per day exposure, hazard quotient (HQ), hazardous index (HI), and incremental lifetime cancer risk (ILCR). The following equations (1, 2, and 3) [21] were used to calculate non-carcinogenic chronic daily intake through dust ingestion, inhalation, and dermal contact. For HQ calculation of each exposure route, Equation (4) was used, and analysis of HI was carried out by combining the HQ of different exposure routes (Equation (5)) [21].
Ingestion dose-nca = Cn × (Ring × EF × ED/ BW × ATnca) × CF(1)
Inhalation dose-nca = Cn × (Rinh × EF × ET × ED/ PEF × BW × ATnca)(2)
Dermal dose-nca = Cn × (SA × SL × ABSd × EF × ED/ BW × ATnca) × CF(3)
HQ = Exposure route- nca/RFD(4)
HI = (HQ-Ingestion + HQ-Inhalation + HQ-Dermal)(5)

Equations (6)–(8) were used to estimate carcinogenic risk exposure via different exposure routes. Moreover, the total carcinogenic risk was evaluated by calculating the combination of all exposure routes and cancer slope factor (SF) in Equation (9) [21].
Ingestion dose-ca = Cn × (IR × EF/ ATnca) × CF(6)
Inhalation dose-ca = Cn × (EF × ET × ED/ PEF × 24 × ATca)(7)
Dermal dose-ca = Cn × (ABSd × EF × DFSadj/ATca) × CF(8)
ILRC = (Ingestion dose-ca × SF oral) + (Inhalation dose-ca × SF inhalation) + (Dermal dose-ca + SF dermal)(9)

Cancer slope factor (SF) (mg/kg/day) was not available for most studied BFRs except for BDE 209. For BDE 209, only oral SF (0.007) was available for oral and dermal routes to calculate ILRC. In the above equations, Cn signifies the concentrations of the BFRs (µg/g) in indoor dust and PM_10_. For the above calculations, the 90th percentile was used. ‘Ing’ indicates dust ingestion rate. For these calculations, children’s high dust intake (200 mg/day) was used due to the prevailing dry arid and dusty conditions in Saudi Arabia throughout the year [20]. In indoor, air conditioning is used by the Saudi public for cooling purpose throughout the year, which results in regular air circulation indoors and, thus, leads to the accumulation of a high quantity of indoor fine dust particles [20]. This is also evident with high levels of PM_10_ found in children’s rooms (Table 1). The Ring represents the inhalation rate (m^3^/day), which was 7.6 for children, as reported in the literature for such calculations [20]. Exposure frequency (EF) was 350 days/year, and the duration of exposure was two years [22,23]. Other parameters are exposed skin area (SA) (1600 cm^3^), dust to skin adherence factor (SL) (0.5 mg/cm^2^) [23], dermal absorption factor (ABSd) (0.03) [22], particle emission factor (PEF) (1.36 × 109 m^3^/kg) [22], body weight (BW) (15 kg) [20], lifetime (LT) (70 years) [20], conversion factor (CF) (1 × 10^–6^) [22], dust dermal contact factor-age-adjusted (DFSadj) (113 mg/day) [23], exposure time (ET) (17.8 hrs/day) [22], average non-carcinogenic exposure time (ATnca) (ED × 365), and average carcinogenic exposure time (ATca) (LT × 365) [22].
Estimated daily intake = (Cn × IR)/BW(10)

In the above Equation (10), Cn signifies the concentrations of the BFRs in indoor dust (ng/g) and PM_10_ (pg/m^3^). In these calculations, the mean and 90th percentile of the concentrations were used to calculate different exposure scenarios. IR represents ingestion and inhalation rate. Low (50 mg/day) and high (200 mg/day) dust intake by children was assumed to calculate low and high-end exposure. Different body weights (BW) for the children’s group were considered (toddlers (12 kg), young kids (6–8 years) 25 kg, and teenagers (40 kg). According to the exposure factor handbook, the inhalation rate was considered as 8.93, 11.96, and 15.17 m^3^ for toddlers, young children, and teenagers [24].

## 3. Results

### 3.1. Levels and Profile of BFRs in Indoor Dust and PM_10_ from Children’s Rooms

#### 3.1.1. Particulate Matter 10 (PM_10_)

The primary statistical summary of measured levels of PM_10_ and BFRs in dust and PM is provided in Table 1. The levels of PM_10_ varied between 15 and 275 µg/m^3^ with a median value of 58 µg/m^3^. PM is a criterion air pollutant which on exposure has adverse health implications [25]. The levels of PM_10_ found in the present study is on the higher side, especially in samples from households above 100 µg/m^3^, which is a cause of concern. Many studies have reported the effects of PM inhalation on the functioning of the respiratory system and negative impacts on the cardiovascular and nervous systems [26,27,28]. This is especially concerning for young children with their rapidly developing bodies. High levels of PM_10_ were found in those children’s rooms with windows towards the busy roads. The outdoor weather conditions also significantly impacted the indoor PM_10_; when the outdoor weather was stormy and dusty, the PM_10_ was high in those samples. 

Most of the PM_10_ samples collected from children rooms showed a lower presence of BFRs (Table 1). BDE 209 analyzed BFRs in PM_10_ samples with a median concentration of 75 pg/m^3^. This indicates that the broader application of BDE 209 still has implications for its occurrence, although its use has been regulated for specified use since 2016 [29]. Most of the other BFRs were present in <50% PM_10_ samples (Table 1, Appendix A. However, TBB was found at an average concentration of 100 pg/m^3^ primarily due to its high presence in two PM_10_ samples from children’s rooms. BDE 209 and TBB were the major BFRs in PM_10_ by contributing 44% and 37%, respectively, based on average concentrations while, among others, BTBPE contributed > 8% based on average levels (Figure 1). However, using median levels, BDE 209 contributed an overwhelming 71%, and BTBPE and BDE 183 were the other essential contributors (Figure 1). All other BDEs and alternative BFRs contributed < 2.5% each (Figure 1). This indicates in some households that high levels of alternative BFRs were found. Although alternative BFRs are not as uniform as BDEs, they are present indoors and replace the regulated PBDEs. Simultaneously, the lower levels of PBDEs showed that their levels are levelling off after the ban on using these chemicals in consumer products. 

#### 3.1.2. Floor Dust 

The total concentration of ∑BFRs in the analyzed dust ranged between 1300 and 61,500 ng/g (Table 1, Appendix A). Among all analyzed BFRs, BDE 209 was present at the highest concentrations of up to 60,800 ng/g and were found in more than 80% of the analyzed samples, while other PBDE congeners were detected at lower concentrations and detection frequencies. TBPH, TBB, and BTBPE were the other important BFRs with median levels of 15, 10, and 8 ng/g, respectively. Most of the PBDE congeners were present in less than 50% of dust samples, although they were found at high concentrations in a few samples. This indicates that some sampled rooms contain items treated with PBDEs which contaminate their room (Table 1 and Figure 2), while in some dust samples, alternative BFRs were present at higher concentrations, indicating that these BFRs are also released from the treated products. All alternative BFRs were present in >60% of dust samples, indicating their higher presence than Octa- and Penta-BDEs (Table 1 and Figure 1). BDE 209, TBB, and TBPH contributed 67.6%, 23.3%, and 5.4 % in the dust BFR profile based on average concentrations (Figure 2), while all other BFRs contributed <1%. However, when the median levels were used to study the profile of BFRs in dust samples, BDE 209 contribution was significantly large, at more than 98% (Figure 2), which indicate its use and persistence in the indoor environment. At the same time, it also suggests that, although BDE 209 is regulated, dust is still an essential source of BDE 209 indoors. All other BFRs contributed <1% each (Figure 2). The skewed distribution (Appendix A) of these chemicals in both dust and PM_10_ indicates that both alternative and regulated BFRs are present in children’s rooms at varying concentrations. This indicates that these chemicals might be affected by various factors such as furniture and toys, old or new, electronics, and the amount of these products. Our results showed that both dust and PM_10_ are sources of high molecular weight BFRs. 

TBB and TBPH are the main components of Firemaster-550, marketed as an alternative to regulated Penta-BDE [1]. These chemicals in both dust and PM_10_ from children’s rooms suggest their use in many products used for children, such as flexible polyurethane foam, which might be utilized to cushion furniture. No positive correlation was observed between TBB and TBPH (*p* > 0.05); this might indicate other sources than FM-550 for these chemicals, such as DP-45, which contains TBPH [15]. There were two samples with high concentrations of TPBH but low levels of TBB. This might indicate the use of DP-45 in insulation cables, wires, and coated fabrics, etc. [30]. 

A two-sample t-test was applied to determine the difference in BFRs between children’s rooms with less (2) and more children (>2), direct and indirect ventilation, electronics and electrical appliances, and new (less than one year) and old toys (older than one year). However, no significant differences were found (*p* > 0.05) in the BFR levels for different parameters. Among many other factors such as sampling (different people collecting the samples) and obtaining similar scale information via questionnaire, the small size of the data set made it difficult to find a statistically significant difference in the collected socioeconomic parameters. More extensive studies are needed for meaningful statistical analysis. Similarly, Spearman rank-order correlation was applied to explore the possibility of familiar sources for BFRs in dust and PM_10_ samples and among different BFRs. However, no significant correlation (*p* > 0.05) was found between levels of different BFRs in dust and PM_10_. Similarly, no significant positive correlation (*p* > 0.05) was found among different BFRs. This may suggest various emission sources and transport mechanisms for BFRs inside the children’s rooms. Another reason might be the different environmental fates of these BFRs after their release indoors.

### 3.2. Comparison with Literature Data

Many studies have reported PBDEs globally in indoor dust and air in the last two decades. However, after regulating against certain PBDEs, focus was also given to the formulation of alternative BFRs and their presence in the environment, especially indoors. Several recent studies have reported PBDEs and alternative BFRs in indoor dust; however, few studies are available on their occurrence in indoor air. Nonetheless, some studies are available in the literature that reported these chemicals indoors, linked explicitly with children. Therefore, the comparison was made with many other studies, mainly published within the last ten years, on their occurrences indoors, such as in households, daycare, and schools (Figure 3, Appendix A). The median levels of BFRs in the current study were higher than those reported from Iraq [31], Egypt [32], Kuwait [10], Pakistan [10], Taiwan [33], Japan [34], Australia [35], Turkey [36], Poland [37], Norway [38], Germany [39], Portugal [40], Sweden [41], and Romania [13] (Figure 3, Appendix A). The levels of PBDE congeners and alternative BFRs were lower than in an earlier reported study on household dust from Jeddah, KSA, except for BDE 209 (Figure 3, Appendix A). The median levels of BDE 209, many times higher than in the previous study, might indicate the release of these chemicals from various items used in children’s rooms, e.g., electronics, upholstery fabric, and other toys where recycled material is used [2]. However, they were lower than those reported from China [42], Korea [43], the USA [44,45,46], and Spain [47] (Figure 3, Appendix A). Most current studies reported BDE-209 as the most dominating BFR in dust samples (Figure 3) except for dust samples from the USA and Norway, where BDE-49, -99, and alternative BFRs also contributed significantly to the profile of BFRs (Figure 3). These variations in dust samples are due to variation in fire safety regulations in the different jurisdictions. Studies have shown that, historically Penta-BDEs were found at high levels in environmental samples from North America. At the same time, Deca-BDEs were present at high concentrations in European and Asian environmental samples [10,13,31,32,33,34,35,36,37,38,39,40,41,42,43,44,45,46,47]. This might also indicate the industrial preference for certain BDE formulation in different countries. These studies have been conducted at various times, during which regulations were put in place to use different commercial formulations of BDEs. Therefore, this might be another reason for variation in the profile of BDEs when comparing different countries. New studies from these countries and the same regions are needed to confirm if BDE levels and profiles have changed after regulating their consumer products.

Unlike dust, few studies have reported BFRs or focused on PBDEs in indoor air. This is primarily due to the ease of dust sampling compared to air sampling. One recent study conducted by Cequier et al. [38] analyzed alternative BFRs in indoor air from Norway but found <dl median levels for TBB, TBPH, and BTBTPE from home and classroom air samples. BDE 47, 99 and 209 were the major congeners in indoor air samples from homes, classrooms, and daycares, reported in the literature from different countries (Appendix A). The levels of PBDEs were much higher in air samples from USA households [48,49], Swedish daycare and homes [50,51], Norwegian households and classrooms [38], and Korean schools, academies, and households [43] than those found in the present study. However, PBDEs were found in a similar range to those reported from Kuwaiti homes [52], Swedish apartments [50], Korean classrooms [53], UK households [54], and Hong Kong and Taiwan homes [55,56]. The difference in the levels of indoor PBDEs in various countries might be attributed to the use of these chemicals according to local fire safety regulations and air sampling methodology, building characteristics, different weather seasons during sampling, and analytical protocols. 

Many other studies on PBDEs in environmental samples from the USA have shown that Penta-BDEs are found at high concentrations [44,45]. This is primarily due to its heavy use in consumer products to fulfil stringent fire safety regulations. However, in other countries, BDE-209 is the most dominant PBDE congener in environmental samples, including indoor dust [13]. This indicates different fire safety regulations among countries. Despite the limitation on the use of PBDE formulations, complete phasing out of PBDEs is still not followed strictly in many regions, and these chemicals return during recycling in new products. Similar scenarios have been discussed by Dirtu et al. [13]; despite the implementation of strict regulations, exposure to PBDEs is likely to continue for some time due to their persistence in the environment and ubiquity in older consumer materials. The high levels of these chemicals in the environment in China, the USA, and other industrial-scale producers and users might indicate their leaching into the surrounding environment. During industrial production, their use and persistence in the environment lead to high PBDEs and new BFRs (Figure 3, Appendix A).

### 3.3. Human Risk Assessment

Several studies have reported that exposure to indoor pollutants is linked with various health conditions. The most important pollutants in the study were BDE 209, which have been reported to have the potential to lead to various animal tumors on a laboratory scale [57]. Many PBDE congeners are reported in human samples such as serum, milk, fat, hair, etc. Studies have suggested positive correlations between exposure to PBDEs and health conditions such as neurobehavioral and reproductive disorders, thyroid hormone disruption, etc. [58,59,60,61]. However, there are no studies available which suggest that PBDEs exhibit carcinogenic potential in humans. With limited evidence of carcinogenicity in humans and animals, the International Agency for Research on Cancer (IARC) and USEPA classified PBDEs as Group 3 and Group D carcinogens (not classifiable as to its carcinogenicity to humans), respectively [58]. Like PBDEs, few studies have focused on studying the impact of new BFRs on human health; therefore, data is limited [1,62]. A recent study from China found a significant correlation between thyroid disruption and new BFRs in serum [63].

Different exposure scenarios were calculated for various exposure routes, using other equations to investigate the health risk associated with long term and daily exposure to BFRs via dust and PM_10_, (1–10). As explained above, many BFRs are linked with multiple health problems; therefore, long term non-carcinogenic risk (HQ and HI), as well as carcinogenic risk, were calculated. ILCR was calculated to look at the potential long-term cancer risk via dust and PM_10_ exposure for Saudi children from exposure in their rooms. As shown in Table 2, the value of HI was <1 for all studied BFRs, indicating a low non-carcinogenic risk to Saudi children from exposure to indoor dust and PM_10_ from their room. The ILRC was collected only for BDE 209 using Equations (6)–(9) because cancer slop factor (SF) values are missing for other BFRs in the literature. Furthermore, even for BDE 209, only SF oral was available in the literature, used for oral and dermal exposure routes. The probabilistic ILCR assessment was 2.05 × 10^−7^ (Table 2) which is well below the USEPA recommended safe limit (1.00 × 10^−4^) for long term cancer risk. This indicates that Saudi children have a low carcinogenic risk linked to BDE 209 from its presence in their rooms.

Using the results for analyzed BFRs in PM_10_ and dust from children’s rooms, various exposure scenarios were calculated using averages, and 90th percentile levels concentrations were calculated for daily exposure (Figure 4A,B, Appendix A). Both the low end and high end daily calculated exposure were many times below the reference dose (RFD) values for all toddlers, young, and teenage children (Figure 4A,B, Appendix A). However, many of these RFD values, especially for new BFRs, need to be updated based on current toxicological studies. In addition, many of these individual BFRs have a similar toxicological impact on health. Therefore, the synergetic effect of these chemicals might cause serious health concerns in the long term to the rapidly developing bodies of young children. However, it needs to be cautioned that these preliminary estimates are based on a small data set. Another significant issue is the lack of updated toxicological studies on many of the chemicals.

Consequently, no RFD or cancer slope factors are available for many BFRs, making it challenging to estimate risk accurately. Updated RfDs, cancer slope factors, and better data on the bioavailability of BFRs are required to improve risk assessments. Therefore, this study has its limitations. Nonetheless, it indicates the likely range of BFR exposure to young Saudi children from their rooms. However, multiple classes of other organic and inorganic pollutants can exert a similar health impact with long term exposure. Therefore, large scale temporal monitoring of these indoor chemicals, significant to children, is warranted to understand the health risk accurately to children’s developing bodies from exposure to various indoor pollutants.

## 4. Conclusions

This is the first study reporting on BFRs in Saudi children’s rooms, an essential microenvironment. The ∑BFR concentrations in children’s room dust were higher than those reported for the region’s household dust. Although BDE 209 was still the major contaminant as in previous study, alternative BFRs were present at much higher levels than Penta and Octa-BDEs in both PM_10_ and dust samples. This indicates that a ban on the use of Penta and Octa-BDEs might be responsible for the lower presence of these chemicals; however, the increasing number of alternative BFRs indoors is concerning. Daily exposure and long term non-carcinogenic and carcinogenic risk were minimal for Saudi children from BFRs inside their rooms. However, this study suggests that children are exposed to chemicals from their rooms and these need to be identified along with other environmental pollutants. This study has some limitations, especially the low number of samples and small number of BFRs analyzed. Therefore, large-scale indoor studies, especially important for young children in daycare and primary schools, are warranted. These large-scale studies are needed to understand BFRs and other indoor chemical pollution dynamics and assess the impact of exposure in the long-term to children of different age groups.

## Figures and Tables

**Figure 1 ijerph-18-06421-f001:**
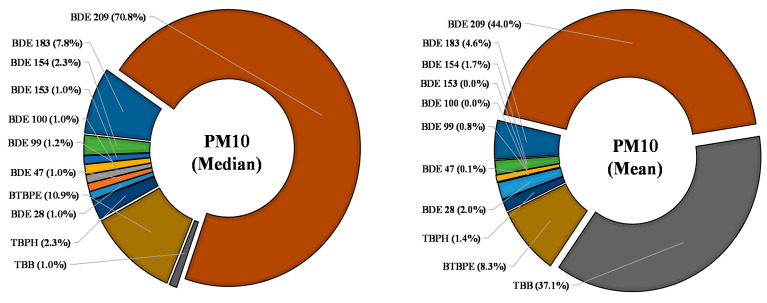
Profile of studied BFRs with median and mean values found in indoor PM_10_ from Saudi children’s room.

**Figure 2 ijerph-18-06421-f002:**
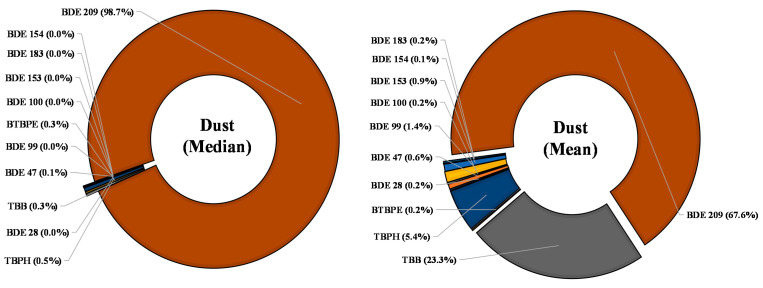
Profile of studied BFRs with median and mean values in indoor dust samples from Saudi children’s rooms.

**Figure 3 ijerph-18-06421-f003:**
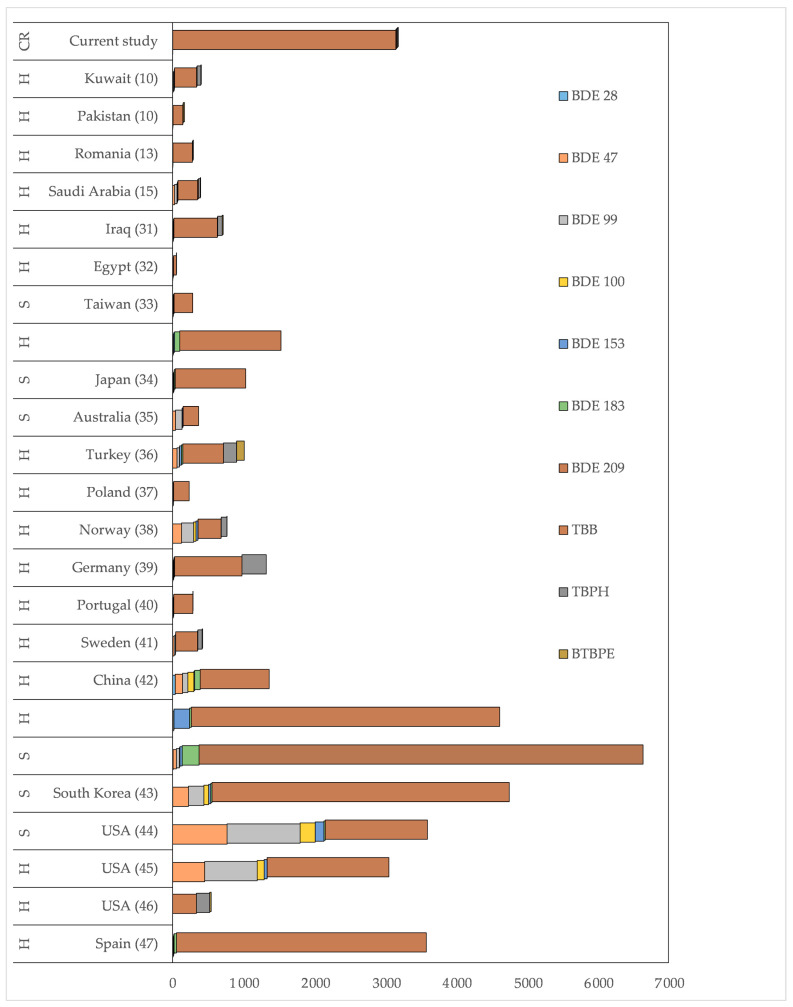
Comparison of median levels (ng/g) of BFR data from different countries for indoor dust. Values on the longitudinal axis are in ng/g. H, S, and CR represent household, school, and children’s rooms, respectively.

**Figure 4 ijerph-18-06421-f004:**
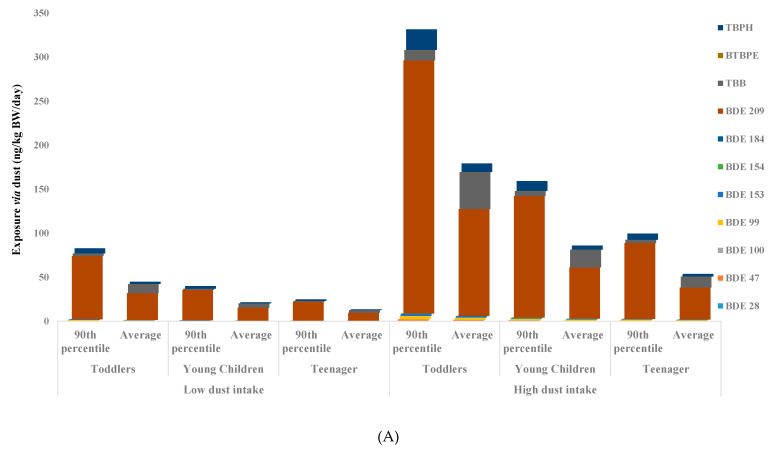
(**A**) Estimated daily exposure (ng/kg/bw/day) to BFRs via dust ingestion for Saudi young children from their rooms. (**B**) Estimated daily exposure (pg/kg/bw/day) to BFRs PM_10_ inhalation for Saudi young children from their rooms.

**Table 1 ijerph-18-06421-t001:** Concentrations (ng/g) of analyzed BFRs in indoor dust (ng/g) and PM_10_ (pg/m^3^) collected from Saudi children room in Jeddah, Saudi Arabia.

Analytes	Dust (ng/g)		Air (pg/m^3^)	
	Mean ± Standard Deviation	Median (Min–Max)	Mean ± Standard Deviation	Median (Min–Max)
BDE 28	15 ± 18	<0.2 (<0.2–95)	5 ± 7	<1 (<1–50)
BDE 47	65 ± 150	2 (<0.2–500)	1 ± 1	<1 (<1–3)
BDE 100	20 ± 60	<0.2 (<0.2–230)	0 ± 0	<1
BDE 99	145 ± 375	<0.2 (<0.2–1650)	2 ± 3	2 (<1–15)
BDE 153	95 ± 290	<0.2 (<0.2–1380)	0 ± 0	<1
BDE 154	10 ± 25	<0.2 (<0.2–115)	5 ± 5	2 (<1–20)
BDE 183	25 ± 100	<0.2 (<0.2–485)	12 ± 15	8 (<1–45)
BDE 209	7270 ± 12,880	3150 (<2–60,800)	120 ± 150	75 (<10–700)
TBB	2500 ± 1140	10 (<2–56,020)	100 ± 495	<2 (<2–2500)
BTBPE	17 ± 65	8 (<1–320)	20 ± 25	11 (<2–90)
TBPH	580 ± 1520	15 (<2–6530)	4 ± 5	2 (<2–15)
∑BFRs	10,900 ± 18,300	3950 (1300–61,500)	300 ± 500	180 (<2–2550)
PM_10_			75 ± 62	58 (15–275)

**Table 2 ijerph-18-06421-t002:** Calculated potential cancer (ILCR) and non-carcinogenic (HQ and HI) risk assessment for Saudi children using 90th percentile values of BFRs in floor dust collected from their rooms.

Non-Carcinogenic	CDI (Ingestion-nca)	CDI (Inhalation-nca)	CDI (Dermal-nca)	HQ-Ingestion	HQ-Inhalation	HQ-Dermal	HI
BDE-28	6.39 × 10^−7^	2.54 × 10^−10^	7.67 × 10^−8^	6.39 × 10^−3^	2.54 × 10^−6^	7.67 × 10^−4^	7.16 × 10^−3^
BDE-47	1.01 × 10^−6^	0.00	1.21 × 10^−7^	1.01 × 10^−2^	0.00	1.21 × 10^−3^	1.13 × 10^−2^
BDE-100	1.21 × 10^−7^	0.00	1.46 × 10^−8^	1.21 × 10^−3^	0.00	1.46 × 10^−4^	1.36 × 10^−3^
BDE-99	2.87 × 10^−6^	2.24 × 10^−11^	3.45 × 10^−7^	2.87 × 10^−2^	2.24 × 10^−7^	3.45 × 10^−3^	3.22 × 10^−2^
BDE-153	1.92 × 10^−6^	0.00	2.31 × 10^−7^	9.61 × 10^−3^	0.00	1.15 × 10^−3^	1.08 × 10^−2^
BDE-154	8.09 × 10^−8^	7.07 × 10^−11^	9.70 × 10^−9^	4.04 × 10^−4^	3.54 × 10^−7^	4.85 × 10^−5^	4.53 × 10^−4^
BDE-183	3.16 × 10^−8^	1.86 × 10^−10^	3.79 × 10^−9^	1.05 × 10^−4^	6.19 × 10^−7^	1.26 × 10^−5^	1.19 × 10^−4^
BDE-209	2.21 × 10^−4^	1.63 × 10^−9^	2.65 × 10^−5^	3.15 × 10^−2^	2.33 × 10^−7^	3.78 × 10^−3^	3.53 × 10^−2^
TBB	9.10 × 10^−6^	5.45 × 10^−11^	1.09 × 10^−6^	4.55 × 10^−4^	2.73 × 10^−9^	5.46 × 10^−5^	5.10 × 10^−4^
BTBPE	1.87 × 10^−7^	3.27 × 10^−10^	2.25 × 10^−8^	7.70 × 10^−7^	1.35 × 10^−9^	9.24 × 10^−8^	8.64 × 10^−7^
TBPH	1.79 × 10^−5^	6.55 × 10^−11^	2.15 × 10^−6^	8.94 × 10^−4^	3.27 × 10^−9^	1.07 × 10^−4^	1.00 × 10^−3^
Carcinogenic	CDI (Ingestion-ca)	CDI (Inhalation-ca)	CDI (Dermal-ca)	ILRC-Ingestion	ILRC-Inhalation	ILRC-Dermal	ILRC
BDE-209	2.67 × 10^−5^	3.84 × 10^−9^	2.57 × 10^−6^	1.87 × 10^−7^		1.80 × 10^−8^	2.05 × 10^−7^

## Data Availability

All the data is presented in the article and associated Appendix A.

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
