# Peer review of "Brominated Flame Retardants in Children’s Room: Concentration, Composition, and Health Risk Assessment"

_ijerph, 2021, doi:10.3390/ijerph18126421_

Round 1
Reviewer 1 Report
Please see the attached file. Thanks a lot.

Author Response
Reviewer #1:
This study investigated the BFRs levels for 30 households in Saudi Arabia and further calculated health risk for children. I suggest that the editor can consider accepting it after major revision. My questions and comments are shown below.
Response: Thanks for your remarks. Necessary changes have been made in the text as per the reviewer comments. We addressed most of the comments in the revised manuscript.
In the line 80, what is the EPA (Environmental Protection Administration?)?
Response: EPA = Environmental Protection Agency, and we explained in the revised manuscript.
How did the authors decide to collect samples for the 30 study households? And, what is the selection criteria? Please describe them in the text.
Response: Thanks for your remarks. This was the first study from the region, so not very strict selection criteria applied. For this study, paired particulate material (PM10) and indoor dust samples were collected from children’s room (N=30) of different households of Jeddah, Saudi Arabia, between August and December 2019. The household with young children with a separate room for children was randomly selected from the general population who voluntarily participated in this study.
In Table 1, please note the full name for STD.
Response: Thanks. STD = standard deviation and explained in the revised manuscript
Please also note the full name for H, S, and CR in figure 3.
Response: Agreed and provided in the revised manuscript
In Figure 3 and 4, the authors completed the comparison for the BFRs levels from previous studies, which means many studies have investigate the BFRs in indoor dust and air. I suggest that authors should describe results in the introduction part and point out the new points in the study.
Response: We have agreed with your remarks. Yet, this information is not the primary goal of this study, and previously available literature/reports included only for comparison purposes. The main focus is to study the BFRs in dust and PM10 collected from Saudi children. If we retain this information in the introduction, it would be lengthy. Correspondingly, we have provided this information in the results and discussion section, which is more suitable for comparison and discussion.
In Figures 1,2, and 4, there are same color for BDE 209 levels. But in figure 3, the authors used different color to present BDE 209. I would suggest using same color to present BDE 209 in all figures.
Response: Agreed, and we changed the figure with a similar colour scheme for BDE 209 in figure 3.
Please subscript for “PM10” in the text.
Response: Thanks. We subscripted PM10 in the revised manuscript.
On the part of health risk assessment, is teer exposure parameters that can be used to calculate health risk for children from different exposure pathway from Saudi Arabi’s population?
Response: Thanks for your remarks. According to EPA (Environmental Protection Agency) guidelines, we used parameters and values found in the literature for health exposure assessment. We could not find any specific parameters for the Saudi population; therefore, we used parameters described in the literature and provided references.
Why is the predominant BFRs compound is not BDE 209 in Norway and US dust, please explain and authors seems to present and compare BFRs concentrations from the previous studies but without much discussion, please discuss it.
Response: Thanks. BDE 209 was still the most dominant BFR in dust samples from Norway, but with significant contribution from Penta-BDEs, as shown in Table S2 and Figure 3. We have provided a new discussion in the revised manuscript on these points.
Do the authors also collected outdoor dust and air samples, if not. Why?
Response: Thanks. We did not collect the outdoor dust and air samples for this study since it was not in the main scope of this study; however, we have a plan for such sampling in the following survey.

Reviewer 2 Report
The paper reports measurements of BFRs in 30 rooms in Saudi Arabia. The paper is of significance but requires improvements to the written english and the clarity of some explanations given. More work should be attempted to understand when and why higher concentrations were detected. More generally the paper would appeal to a wider audiance if some more general explanation was given in the introduction and conclusions. Some specific comments to address:
- Table 1 reports and average and median value. Is the average here the mean? If so it should use mean rather than average.
- The indication from table 1 is that median concentrations in both dust and air are very low. If this is the case the average is being pushed up by some higher values. Nowhere in the publication or supplementary materials can we see the range in measured concentrations across different rooms. It is important to establish how these are distributed. If some apartments have very low concentrations, others very high, then this should be discussed.
- Very few details of the sampled rooms are given. Other than the fact they are in Jeddah. Are the rooms in appartments? how are they ventilated? Are they in seperate dwellings/buildings or the same
- Ventilation is recorded for the rooms. Whilst some factors, are explored via t-tests, others are not. Even if no significance is found they this should be reported.
- This is important as it is one thing to report concentrations and compare to other studies. However, the important next step is to understand drivers for higher concentrations. If this could not be established through this study, what would be needed in future work to do so.
- The paper refers to children's rooms. It is assumed this means bedrooms but it would be better if the manuscript directly refered to them as bedrooms to avoid any ambiguity.
- When was the sampling conducted. Is there likely to be any bias in sampling at particular points in the year (e.g. under higher or lower ventilation rates or lower/higher temperatures.
- Avoid the use of first person, e.g. 'we' in the document.
- The english and clarity of explanation needs to be improved throughout.
- Figure 1 and figure 2 are not intuitively clear. This is the respective proportion of BFRs as means and medians? What is this as a share of the total PM10 collected?
- Perhaps figures 1 and figure 2 could be broken down per room to better describe the variation mentioned earlier.
- Values of previous studies are included. Do these studies not indicate what factors might be significant in higher concentrations?
- Many plots have small formatting issues.
- The 'H', 'S' and 'CR' in figure 3 should be explained within the legend or caption.
Author Response
Reviewer #2:
The paper reports measurements of BFRs in 30 rooms in Saudi Arabia. The paper is of significance but requires improvements to the written English and the clarity of some explanations given. More work should be attempted to understand when and why higher concentrations were detected. More generally the paper would appeal to a wider audience if some more general explanation was given in the introduction and conclusions. Some specific comments to address:
Response: Thanks for your remarks. Necessary changes have been made in the text as per the reviewer comments. We addressed most of the comments in the revised manuscript. In the revised manuscript tried to provide more details to make it clear.
Table 1 reports and average and median value. Is the average here the mean? If so it should use mean rather than average.
Response: Thanks. Agreed and changed as suggested.
The indication from table 1 is that median concentrations in both dust and air are very low. If this is the case the average is being pushed up by some higher values. Nowhere in the publication or supplementary materials can we see the range in measured concentrations across different rooms. It is important to establish how these are distributed. If some apartments have very low concentrations, others very high, then this should be discussed.
Response: Thanks for your remarks. The skewed distribution of these chemicals in dust and PM10 indicates that both alternative and regulated BFRs are present in children's rooms at varying concentrations. This suggests that these chemicals in the children room might be affected by various products/factors such as furniture and toys, old or new furniture, electronics etc.
Very few details of the sampled rooms are given. Other than the fact they are in Jeddah. Are the rooms in apartments? How are they ventilated? Are they in separate dwellings/buildings or the same?
Response: Thanks. All this information are provided in detail in the supplementary information (please see Table S1). We also mentioned this in the subchapter entitled “Sampling”.
Ventilation is recorded for the rooms. Whilst some factors, are explored via t-tests, others are not. Even if no significance is found they this should be reported.
Response: Thanks. Agreed, and we revealed the important ones with enough data for comparison; we mentioned this in the revised manuscript.
This is important as it is one thing to report concentrations and compare to other studies. However, the important next step is to understand drivers for higher concentrations. If this could not be established through this study, what would be needed in future work to do so?
Response: Thanks for your remarks. Among many other factors, such as sampling (different people collected samples) and getting similar scale information on the questionnaire, the small size of the data set made it difficult to find statistically significant differences for the collected socioeconomic parameters. More extensive studies are needed for meaningful statistical analysis. We will do it in the following survey/study.
The paper refers to children's rooms. It is assumed this means bedrooms but it would be better if the manuscript directly referred to them as bedrooms to avoid any ambiguity. When was the sampling conducted? Is there likely to be any bias in sampling at particular points in the year (e.g. under higher or lower ventilation rates or lower/higher temperatures.
Response: Thanks. We have mentioned in the revised manuscript pointing that: “For this study, we collected paired particulate material (PM10) and indoor dust samples simultaneously from children rooms (N=30) of different selected households of Jeddah, Saudi Arabia between August - December 2019. During the sampling campaign, paired-samples (dust) and air (PM 10) were collected from each household's children's room. Weather in Jeddah is generally dry, dusty, and hot throughout the year, and in-door temperature varied between 18-25 Ċ, due to air conditioning, depending on the household preference. Therefore, sampling from a different time of the year would not make a big difference.”
Avoid the use of first person, e.g. 'we' in the document.
Response: Thanks. Agreed and removed from the revised manuscript.
The English and clarity of explanation needs to be improved throughout.
Response: Thanks. The manuscript is thoroughly improved. Typographic mistakes and other errors removed. The revised manuscript is enhanced with the help of the reviewer’s comments.
Figure 1 and figure 2 are not intuitively clear. This is the respective proportion of BFRs as means and medians? What is this as a share of the total PM10 collected?
Response: Figure 1 is for BFRs in PM 10 (mean and median), while Figure 2 is for BFRs in dust (mean and median) samples. The total share of PM10 is not relevant since we are discussing chemicals, but we have discussed the amount of PM10 in children rooms in the text.
Perhaps figures 1 and figure 2 could be broken down per room to better describe the variation mentioned earlier.
Response: Thanks. Agreed that figures with individual samples would be interesting, but Figures 1 and 2 also describe the variations in data; that is why we have mentioned both mean and median values here. In the revised manuscript, we have included suggested figures in supplementary information since we believe Figure 1 and 2 explain well the variations of BFRs.
Values of previous studies are included. Do these studies not indicate what factors might be significant in higher concentrations?
Response: Thanks. We have added a new discussion in this subchapter.
Many plots have small formatting issues.
Response: Typographic mistakes and other errors removed in the revised manuscript
The 'H', 'S' and 'CR' in figure 3 should be explained within the legend or caption.
Response: Thanks for your inputs. Agreed and explained them in the revised manuscript.

Round 2
Reviewer 1 Report
I have only one comment left to the authors. They can describe the study limitation if it is possible. Thanks.
Author Response
Reviewer # 1 (Round 2):
I have only one comment left to the authors. They can describe the study limitation if it is possible.
Response: Thanks for your remarks. To clarify this point, we have mentioned the following statements in the conclusion section.
New text: This study has some limitations, especially the low number of samples and a small number of BFRs analyzed. Therefore, large-scale studies from the indoors, especially important for young children, such as daycares and primary schools, are warranted. These large-scale studies are needed to understand BFRs and other indoor chemical pollution dynamics and assess the impact of exposure on the long-term to children of different age groups.
